# Origin and Evolution of the Human Bcl2-Associated Athanogene-1 (BAG-1)

**DOI:** 10.3390/ijms21249701

**Published:** 2020-12-18

**Authors:** Peter Nguyen, Kyle Hess, Larissa Smulders, Dat Le, Carolina Briseno, Christina M. Chavez, Nikolas Nikolaidis

**Affiliations:** 1Center for Applied Biotechnology Studies, and Center for Computational and Applied Mathematics, Department of Biological Science, College of Natural Sciences and Mathematics, California State University Fullerton, Fullerton, CA 92834-6850, USA; peter_nguyen@Fullerton.edu (P.N.); l.smulders@csu.fullerton.edu (L.S.); datqle@yahoo.com (D.L.); cbriseno89@csu.fullerton.edu (C.B.); christinachavez@csu.fullerton.edu (C.M.C.); 2Department of Genome Sciences, Molecular and Cellular Biology Graduate Program, University of Washington, Seattle, WA 98195, USA; kylehess@uw.edu

**Keywords:** Hsp70 co-chaperones, single nucleotide polymorphisms, chaperon function, natural selection

## Abstract

Molecular chaperones, particularly the 70-kDa heat shock proteins (Hsp70s), are key orchestrators of the cellular stress response. To perform their critical functions, Hsp70s require the presence of specific co-chaperones, which include nucleotide exchange factors containing the BCL2-associated athanogene (BAG) domain. BAG-1 is one of these proteins that function in a wide range of cellular processes, including apoptosis, protein refolding, and degradation, as well as tumorigenesis. However, the origin of BAG-1 proteins and their evolution between and within species are mostly uncharacterized. This report investigated the macro- and micro-evolution of BAG-1 using orthologous sequences and single nucleotide polymorphisms (SNPs) to elucidate the evolution and understand how natural variation affects the cellular stress response. We first collected and analyzed several BAG-1 sequences across animals, plants, and fungi; mapped intron positions and phases; reconstructed phylogeny; and analyzed protein characteristics. These data indicated that BAG-1 originated before the animals, plants, and fungi split, yet most extant fungal species have lost BAG-1. Furthermore, although BAG-1’s structure has remained relatively conserved, kingdom-specific conserved differences exist at sites of known function, suggesting functional specialization within each kingdom. We then analyzed SNPs from the 1000 genomes database to determine the evolutionary patterns within humans. These analyses revealed that the SNP density is unequally distributed within the *BAG1* gene, and the ratio of non-synonymous/synonymous SNPs is significantly higher than 1 in the BAG domain region, which is an indication of positive selection. To further explore this notion, we performed several biochemical assays and found that only one out of five mutations tested altered the major co-chaperone properties of BAG-1. These data collectively suggest that although the co-chaperone functions of BAG-1 are highly conserved and can probably tolerate several radical mutations, BAG-1 might have acquired specialized and potentially unexplored functions during the evolutionary process.

## 1. Introduction

All living species cope with their ever-changing environments by activating the cellular stress response (CSR) [1]. The major components of the CSR are a set of proteins named heat shock proteins (HSPs). Several HSPs are molecular chaperones, playing a pivotal role in protein homeostasis under pathological and stressful conditions by supporting protein folding and refolding, and protein transportation across various membranes [2]. These proteins also inhibit apoptotic pathways and promote various immune responses by recruiting cytokines and anti-inflammatory effectors [3], and have been associated with several diseases, including cancer, cardiovascular, and neurodegenerative diseases [4,5,6]. In addition to their indispensable intracellular functions, several HSPs are also found at the plasma membrane [7,8,9] and the extracellular environment, where they function in cell signaling and immunity [10,11,12]. HSPs are diverse in function, structure, and domain organization and are usually separated into families according to their molecular size [2].

Evolutionarily, the different HSP families are present in all kingdoms of life, i.e., bacteria, archaea, and eukaryotes [13,14]. Furthermore, when the members of each family, e.g., Hsp70s, are compared to each other, they exhibit a remarkable amount of conservation at their function, structure, and primary sequence. Therefore, the HSP system’s main components are conserved and, in most cases, stem from the same ancestral molecule [13,14]. Such a high degree of conservation has long been attributed to the process of natural selection in the form of purifying selection [15,16].

Although the main HSP components are present and highly conserved in all species, eukaryotes have significantly diversified several of them, including Hsp70s, with eukaryotic genomes coding 10 to 15 different Hsp70s compared to the one to three Hsp70s coded within bacterial genomes. This gene family diversification has long been attributed to gene or genome duplication events that resulted in homologous proteins with specialized functions [15]. For example, there are specialized Hsp70 systems in the mitochondria, endoplasmic reticulum, cytosol, and plant plastids. Similar levels of conservation and gene multiplication hold true for almost all other HSP families, including Hsp40s.

Therefore, common ancestry (homology) explains the presence of the Hsp70 system in all species, and gene duplication explains its amplification in different eukaryotic phyla. However, in several cases, the system “reinvented” itself even though this was not the most parsimonious solution to the cell stress response problem. For example, the bacterial Hsp70 system comprises three major proteins, i.e., Hsp70 (DNAK), Hsp40 (DNAJ), and GrpE [17,18]. In this system, the DNAK is the major chaperone, responsible for protein folding and unfolding; DNAJ is a co-chaperone, responsible for stimulating DNAK’s ATP hydrolysis; and GrpE, the nucleotide exchange factor (NEF), responsible for removing ADP from DNAK. In all eukaryotes, the same homologous system is present only in the mitochondria and the plant plastids [19,20,21].

In contrast, in all the other eukaryotic Hsp70 systems, the Hsp70 and Hsp40 are homologous to their bacterial counterparts, but the NEFs are not [3,22]. Indeed, different eukaryotic species and different Hsp70s use analogous proteins to remove ADP from the Hsp70. Some use a modified version of an Hsp70, named Hsp110 [23], while others use proteins that contain the BAG (Bcl2-associated athanogene) domain [24]. All these proteins bind to the same region on the Hsp70 molecule, causing similar molecular shifts to release the ADP from the Hsp70s’ nucleotide-binding groove. To make things more fascinating, the GrpE and the BAG domains have similar structures and cause identical structural changes upon binding to the Hsp70 polypeptide [25].

These observations reveal that during the evolution of the Hsp70 systems, there were many cases of divergent evolution, in which gene duplication resulted in homologous proteins and structures, and several cases of convergent evolution, in which similar proteins, structures, and functions evolved independently in different eukaryotic lineages. The presence or absence of particular NEFs in diverse eukaryotic lineages raises fundamental questions regarding their origin, evolution, and function. For example, based on the continuous presence of GrpE in all bacteria and mitochondria, we can safely assume that GrpE evolved very early in evolution, predating the origin of the eukaryotic cell, while Hsp110s probably represent eukaryotic innovations.

The case of the BAG domain-containing proteins, however, is more complicated because they are known to exist in vertebrates and plants [26], while their presence in non-vertebrate animals and fungi remains elusive. Additionally, the animal and plant sequences have highly diverged, making the identification and retrieval of plant sequences using, for example, human BAGs, uncertain and error prone [26]. The sporadic presence of BAG-containing proteins in eukaryotes makes their origin obscure, and the question as to whether vertebrate and plant BAGs are homologous or analogous warrants a definite answer. Furthermore, considering that the function of NEFs is relatively straightforward and conserved, it is currently unclear which NEFs evolved to provide redundancy or functional specialization to Hsp70s and potentially provide organisms with selective advantages in particular environments.

In contrast to some research on the interspecies evolution of BAG domain-containing proteins, their evolution and natural variation within a species, including humans, remains unknown. In humans, the most common form of genetic variation is a single nucleotide polymorphism (SNP), a single nucleotide difference between individuals at particular positions within their DNA [27]. The majority of complex traits and diseases in humans are hypothesized to arise from rare SNPs in specific combination with common SNPs. However, how SNPs within BAG encoding genes manifest their functional consequences and whether they are subject to purifying selection or other evolutionary mechanisms within humans remain unknown.

In humans, there are five proteins that contain the BAG domain (BAG-1 to -5) [22]. Among them *BAG1*, one of the most studied BAG proteins, encodes at least three protein isoforms, i.e., BAG-1L, BAG-1M, and BAG-1S [28,29]. All BAG-1 isoforms share two major domains, an N-terminal ubiquitin-like domain (UBQ-like) and a C-terminal BAG domain [30,31]. Although BAG-1′s exact mechanism of action is not entirely clear, human BAG-1 participates in a wide variety of cellular processes, including apoptosis, cell growth and survival, transcriptional regulation, protein refolding and degradation, and tumorigenesis [24,31,32,33].

To answer some of these questions and shed light on the complex evolution of NEFs and, in particular, BAGs, we investigated the origin and evolution of BAG-1 between species, examined the patterns of extant natural variation in humans, and assessed the functional consequences of specific natural polymorphisms in humans.

## 2. Results

### 2.1. Origin and Evolution of BAG-1

#### 2.1.1. BLAST Data Analysis

The human (BAG-1M) and *Arabidopsis* BAG-1 protein sequences contain two domains, the N-terminal ubiquitin-like domain (UBQ-like) and the C-terminal BAG domain, covering approximately 80% of the molecule. Therefore, the first question we aimed to answer was when this particular domain combination arose in evolution. The initial BLAST [34] searches in non-eukaryotic species revealed no BLAST hits with an *e*-value below one could be identified.

We next sought to determine the origin of BAG-1 in eukaryotes. BLAST searches with the human BAG-1 as a query sequence identified sequences with more than 80% coverage of the query sequence and relatively low *e*-values (ranging from 10^−6^–10^−166^) in all available vertebrate genomes. The low *e*-values (which strongly suggest not random sequence alignments), the high query sequence coverage (>85%), and the presence of both functional domains strongly suggest homology. Furthermore, the finding that all vertebrates contain a single BAG-1 strongly implies that BAG-1 is a single copy gene in these species. When the search was expanded in non-vertebrate animals, including choanoflagellates (the presumed unicellular ancestor of multicellular animals), the BLAST hits had relatively high *e*-values (which suggest higher probability of random alignments), only a few had >80% query coverage, were quite sporadic, and did not follow the established taxonomic order. Further examination of the identified sequences using the conserved domains database (CDD) [35] showed that lancelets, sea urchins, several gastropods, nematodes, and a few insects contained both the UBQ-like and the BAG domains. In contrast, in other animals, including choanoflagellates, the identified proteins had only the UBQ-like domain. However, in choanoflagellates, the UBQ-like domain is most similar to the one found in human BAG-1; thus, either these sequences lost the BAG domain, or its sequence diverged beyond recognition by alignment methods. These findings imply that BAG-1 originated before the divergence of the major animal taxa, and it was subsequently lost in multiple animal lineages, while its presence in vertebrates remained continuous.

In contrast to the relatively easy identification of BAG-1 homologs in animals, when fungal and plant genomes were queried using the human sequence, the BLAST results had high *e*-values, less than 80% query sequence coverage, and sequence identities in the borderline of homology (20–25%). These results changed radically when the *Arabidopsis* BAG-1 sequence was used in plants. These searches produced alignments with very low *e*-values, high query coverage, and high sequence identities. These findings, combined with the presence of both domains, strongly suggest the continuous presence of BAG-1 in all eudicots, monocots, and mosses and imply that BAG-1 originated before the split of the major plant lineages.

In fungi, the use of *Arabidopsis* BAG-1 or its closest homolog BAG-4 sequences as queries resulted in very few hits, none of which had the desired characteristics. Similarly, using a reciprocal BLAST strategy, in which the queries and subjects of a search are reversed, resulted in no significant hits neither in fungi nor in animals. However, Saito et al. [36] identified a protein that contained a BAG domain (BAG101) as an interaction partner of the proteasome in *Schizosaccharomyces pombe*. We collected this sequence and determined that it includes both functional domains. When this sequence was used as a query and the fungal protein database as the subject, the alignments produced were of high quality, had low *e*-values, and high query coverage. However, the alignments’ pairwise identity dropped very fast outside the *Schizosaccharomyces* genus, revealing very low amino acid conservation. Domain organization analysis using the CDD further determined that most (but not all) of these sequences contained both the UBQ-like and the BAG domains, and taxonomic analysis revealed that most sequences belonged to Ascomycota and only a few in Basidiomycota, while BAG-1 was absent from both Chytridiomycota and Zygomycota. This irregular and radical pattern suggests that the fungal version of BAG-1 originated before the split of the major fungal lineages, remained single, and subsequently was lost from the majority (>90%) of fungi.

Using the parsimony criterion, the search for the origin of BAG-1 protein in eukaryotes revealed that it appeared early before the split of the lineages that resulted in the different animal, plant, and fungal species (Figure 1). However, these results fail to determine whether the protein originated once in the common ancestor of all eukaryotes or originated multiple (at least three) times in the individual kingdoms’ ancestor. Although the presence of both domains in all species suggests homology and that BAG-1 emerged once in evolution, the very low amino acid identity, which is in the “homology twilight zone” (15–25%), does not allow us to conclude homology. Therefore, we generated multiple sequence alignments and used them to generate phylogenetic trees to resolve the origin issue.

#### 2.1.2. Phylogenetic Trees and Orthologous Relationships between BAG-1 Homologs

To validate orthology and determine evolutionary relationships between collected BAG-1 homologs, we generated phylum-specific phylogenetic trees. The multiple sequence alignments combined with the phylogenetic trees solidified the homology and orthology of BAG-1 proteins in animals, fungi, and plants in agreement with the BLAST results. However, when we used these sequences in combination, the resulting trees had low bootstrap support in most clades, except for the major ones corresponding to the animal, fungal, and plant lineages. Because of this, we selected only a few representative sequences from each major clade and redid this analysis (Figure 2 and Figure 3). The resulting alignment revealed that these BAG-1 orthologs share specific amino acids and are highly divergent (Figure 2). On the other hand, the resultant phylogenetic tree highly supported the expected clustering of BAG-1 proteins from animal, plant, and fungi into separate phylogenetic clades (Figure 3).

Although the specific conserved amino acids and the phylogenetic reconstruction of the species’ established evolutionary relationships strongly suggest that the sequences used are homologs, they do not unequivocally prove this notion. Therefore, we used two different approaches taking advantage of the genomic organization of the *BAG1* genes.

#### 2.1.3. Conservation of Genomic Organization within Vertebrates

In an attempt to identify commonalities between *BAG1* homologs, we examined the genomic organization of genes surrounding the *BAG1* locus in animals, plants, and fungi (Figure 4). Within animals, and specifically within mammals, the genomic organization of the *BAG1* locus was 100% conserved, with *NFX1*, *CHMP5*, *SPINK4*, and *B4GALT1* existing in the same position around *BAG1* with identical transcription orientations. However, syntenic relationships between genes surrounding the *BAG1* locus were lost outside mammals, except for *CHMP5*, a gene located next to *BAG1* in most vertebrate genomes (Figure 4).

Similar results were observed in plants and a few closely related fungal species, but there were no common genes between animals, plants, and fungal species. Therefore, this analysis did not confer much to solve the overall homology issue.

#### 2.1.4. Conservation of Intron Position and Phase

To continue delineating the evolutionary relationships between BAG-1 homologs, we analyzed the introns’ position and phase in each *BAG1* gene (Figure 5). These data first revealed kingdom-specific differences in the number of introns present within *BAG1* homologs. The majority of animal *BAG1* containing six introns, plant *BAG1* containing three introns, and, expectedly, fungal *BAG1* lacked introns entirely. Only one intron, intron four in animal *BAG1* and intron two in plant *BAG1*, was conserved between plants and animals. This intron resides in the same position within the animal and plant *BAG1*. This finding is a strong indicator of homology between the different BAG-1 proteins from animals and plants. However, it does not verify the overall homology, especially in the case of fungi. Therefore, we analyzed further the amino acid sequences to identify conserved (and divergent) patterns of amino acid conservation (or replacement) that could provide further evidence for homology.

#### 2.1.5. BAG-1 Amino Acid Sequence Conservation

Our earlier analysis examining sequence similarity within BAG-1 homologs also revealed a notable divergence between—and, in some instances, even within—kingdoms (Figure 2). This lack of conservation is apparent when analyzing sequence similarity within the UBQ-like and BAG domains. Long stretches of divergent sequences interrupt short stretches of two or three conserved amino acids. In the instances where there were conserved amino acids, they were either 100% conserved or had 100% similarity between BAG-1 homologs. Unsurprisingly, most of these conserved amino acids are required for interacting with Hsp70 (Figure 6).

In addition to the crucial functional amino acids conserved across BAG-1 homologs, several amino acid positions were highly conserved only within specific kingdoms (Figure 7). These conserved differences may represent signatures of functional divergence or functional specialization of BAG-1 homologs within each kingdom.

Structural comparisons between BAG-1 homologs from *H. sapiens*, *H. vulgaris*, *A. thaliana*, *S. pombe*, and *H. marmoreus* revealed that, although these proteins are highly divergent, their BAG domain remains structurally conserved (Appendix A).

These results collectively suggest that BAG-1 proteins in animals, plants, and fungi are homologs and support the notion that the gene originated once in eukaryotic evolution before the split of the major kingdoms and then followed a pattern of divergent evolution. Furthermore, the lack of significant sequence conservation (except for the Hsp70-interaction sites) and the presence of specific conserved changes between the different phyla imply that, although purifying selection preserved the major amino acids, the remaining molecule changed radically. Whether this pattern of evolution is observed within a single species (between different populations) was the subject of the next part of this project.

### 2.2. Sequence Diversity within BAG1 in Humans

#### 2.2.1. Description of the SNP Distribution of BAG1

To determine if BAG1’s evolution in humans follows its interspecies evolution, we analyzed minor allele frequency (MAF), SNP type, SNP distribution, and SNP density (SNPs/1000 bp) within *BAG1* from individual genomes sequenced by the 1000 Genomes project (Table 1). There were 561 SNPs within *BAG1* and surrounding regulatory regions spanning 22,283 bp, amounting to an average SNP density of 25.18 SNPs/1000 bp. Of these SNPs, 227 were found in regulatory regions (22.70 SNPs/1000 bp), and 334 within *BAG1*′s genic region (27.19 SNPs/1000 bp). Out of 334 SNPs present within *BAG1*, 219 were found within introns (26.02 SNPs/1000 bp), 2 within the 5′ untranslated region (UTR) (22.99 SNPs/1000 bp), 76 within the 3′UTR (27.74 SNPs/1000 bp), and 37 within the coding region (35.65 SNPs/1000 bp). Non-synonymous SNPs accounted for 25 of the 37 coding SNPs (31.65 SNPs/1000 bp), with the remaining 12 coding SNPs being synonymous (48.95 SNPs/1000 bp).

Of the 561 SNPs found within *BAG1*, 497 (88.6%) occurred at low frequencies or were rare (defined here as SNPs with an MAF less than 1%) (Table 2). Broken up by region, 199 of the 227 regulatory SNPs (87.7%) and 298 of the 334 genic SNPs (89.2%) were rare. A large proportion of coding SNPs were rare (35 out of 37; 94.6%), which included 12 out of 12 synonymous SNPs (sSNPs) (100%), and 23 out of 25 non-synonymous SNPs (nsSNPs) (92%). Of the 23 rare nsSNPs, 7 (28%) were predicted to be deleterious or damaging to protein function compared to the 2 common nsSNPs that were predicted to be benign or neutral.

We next tested whether the *BAG1* locus has a unique SNP distribution pattern compared to other proximal loci. For this analysis, the SNPs of two loci that reside on each side of the *BAG1* were collected and analyzed. Together with the SNP frequencies and distribution, these results strongly suggest that *BAG1* follows the evolution of the genomic region it resides in (Figure 8).

#### 2.2.2. Selective Pressure of BAG1 in Humans

The next set of analyses aimed to identify regions that experienced different selective pressures. To do this, we analyzed the distribution of non-synonymous (nsSNPs) and synonymous (sSNPs) within domain-coding sequences, i.e., the sequences coding for the UBQ-like and the BAG, and non-domain-coding sequences (Table 3).

Out of BAG1’s 37 coding SNPs (cSNPs), 24 occurred within non-domain regions (42.78 SNPs/1000 bp), while the remaining 13 coding SNPs occurred within the UBQ-like and the BAG (27.43 SNPs/1000 bp). Of these 13 coding SNPs, 6 were found within the UBQ-like (25.97 SNPs/1000 bp), and 7 were found within the BAG (28.81 SNPs/1000 bp).

The distributions of nsSNP and sSNP densities within *BAG1* were also unequal (Figure 9). For sSNPs, eight occurred within non-domain regions (56.74 SNPs/1000 bp), while four occurred within the two domains (38.52 SNPs/1000 bp). Of these four sSNPs, three were found within the UBQ-like (56.79 SNPs/1000 bp) and one within the BAG (19.61 SNPs/1000 bp). For nsSNPs, 16 occurred within sequences lacking a known domain (38.10 SNPs/1000 bp), while nine occurred within sequences coding for the UBQ-like and the BAG domains (24.31 SNPs/1000 bp). Out of these nine SNPs, three were found within the UBQ-like (16.84 SNPs/1000 bp), and six within the BAG domain (31.25 SNPs/1000 bp).

Furthermore, BAG1 contained a lower proportion of nsSNPs than sSNPs, with a pN/pS ratio of 0.65 (Table 4). Domain coding regions had a lower ratio of nsSNPs/sSNPs than non-domain coding regions, with a pN/pS ratio of 0.63. The UBQ-like coding region contains a particularly low amount of nsSNPs, with a dN/dS ratio of 0.3. On the other hand, the BAG domain region is enriched with nsSNPs, with a dN/dS ratio of 1.59 (which is significantly higher than expected by chance; *p*-value < 0.001).

These results support the notion that part of the BAG1 gene might have an accelerated rate of amino acid replacement mutations that could lead to functional changes. This finding agrees with the observations using data from interspecies comparisons and could be explained by either positive selection, relaxation of purifying selection, or at least in the case of humans in the lack of sufficient time for selection to “eliminate” these mutations.

Lastly, to identify SNPs that may alter BAG-1’s function, we analyzed the consequences of nsSNPs in codons encoding known functional amino acids. Out of 25 nsSNPs present within BAG1, three occurred within codons corresponding to known functional amino acids. These three nsSNPs were found to be rare within humans and occurred at highly conserved positions within BAG-1. Out of the three nsSNPs, two occurred within the BAG domain and are predicted to significantly alter BAG-1 function and potentially BAG-1’s interaction with Hsp70.

### 2.3. Functional Characterization of Naturally Occurring Mutations on BAG1

#### 2.3.1. Prioritization of the Mutations and Generation of Recombinant Proteins

The results from the SNP analyses suggested that some of these mutations may affect the function of BAG-1. Therefore, we sought to investigate these notions by using specific biochemical experiments. We first used particular criteria (see the materials and methods section for details and also [15]) to prioritize the non-synonymous SNPs found on the *BAG1* gene and determine which mutations would be experimentally tested (Table 5 and Figure 10).

To experimentally test the effects of these mutations, we first cloned the BAG-1S gene sequence in a mammalian and a bacterial expression vector. Then, we performed site-directed mutagenesis and eventually purified recombinant proteins corresponding to the wild-type BAG-1 and its mutated variants (Figure 11 and Appendix A).

#### 2.3.2. Thermal Shift Assay Using Recombinant Proteins

The purpose of the thermal shift assay was to test whether the presence of a particular mutation alters BAG-1’s protein stability. This assay measures the emitted fluorescence of a fluorophore that binds to hydrophobic protein stretches in sequentially increased temperatures. The results can be used to calculate the melting temperature (Tm) of the protein, thus providing some information on its stability in an aqueous solution. The results of this set of experiments revealed that the mutations generated did not significantly alter the Tm of the resultant proteins as compared to the WT BAG-1 protein (Figure 12 and Table 6), except for the E229K mutation that reduced the predicted Tm by approximately 1 °C. These findings suggest that the mutated variants of BAG-1 have similar structural characteristics with the WT protein.

#### 2.3.3. ATPase Assay Using Recombinant Proteins

The rationale of this assay is based on several publications [33,39,40,41] showing that the presence of BAG-1 will change the rate of ATP hydrolysis of Hsp70s in a concentration-dependent manner. However, the precise mechanism is not yet elucidated. Specifically, the expectation was that when BAG-1 is present in a 0.5:1 (mol:mol) with the HSPA1A, it will inhibit ATP hydrolysis, while when it is present in 2:1 (mol:mol), it will promote ATP hydrolysis. Therefore, in these experiments, we tested whether a particular mutation alters the inhibitory or promoting effect that BAG-1 has on the ATPase activity of HSPA1A.

The results of these experiments are summarized below (Figure 13). (1) The addition of 0.5 μM BAG-1 WT or mutated variants resulted in less released phosphate than the HSPA1A control for all the time points. However, at the 90-min time point variant, E229K resulted in a similar phosphate release as the HSPA1A control (Figure 13A). (2) Suppose we assume that the presence of BAG-1 at 0.5 μM inhibits the ATPase activity of HSPA1A (0–30 and 0–60 data in Figure 13A). In this case, the amount of inhibition is not the same between the different mutations, with some having a more profound effect than others. (3) The inhibitory effect is less pronounced at the 90-min time point for all but the M215V mutation, suggesting altered kinetics (Figure 13A). (4) The addition of 2 μM BAG-1 resulted in increased released phosphate by all mutations at all time points (Figure 13B). (5) All mutations showed an almost linear increase of released phosphate with time, similar to the HSPA1A control. However, at the 90-min time, the WT BAG-1 and the variant K216E proteins resulted in a much higher increase (almost five times more than the HSPA1A alone control) of released phosphate (Figure 13B). These results suggest that the different mutations affect HSPA1A’s function in similar but not identical ways. Furthermore, these data indicate that BAG-1 does not significantly alter the ATPase activity but rather involves this function’s rate (kinetics).

#### 2.3.4. Intracellular Luciferase Protein Refolding

To provide some information on the effect of the mutations in a cellular environment, we used a mammalian expression vector and expressed the WT and mutated proteins (in a GFP-tagged version) in HeLa cells. In these cells, we also expressed firefly luciferase and GFP-tagged HSPA1A.

The intracellular refolding assay is based on the concept that heat-inactivated luciferase cannot refold by itself (and the concentration of native Hsp70s is not high enough to do so). Still, it can refold in the presence of an overexpressed Hsp70. Based on the current literature, the expectation of this assay was that the presence of excess functional BAG-1 would result in loss (inhibition) of the refolding activity of Hsp70 because the latter protein is locked in an “open” conformation (which has low affinity for unfolded proteins). Therefore, if a particular mutation altered the way BAG-1 interacts with Hsp70, it would show different refolded luciferase patterns. The results of this assay (Figure 14) revealed that (1) overexpression of HSPA1A refolds heat-denatured luciferase as expected. (2) Overexpression of BAG-1 alone does not result in luciferase refolding, at least not different from the naturally occurring refolding similar to the empty GFP vector (GFP). (3) Overexpression of both WT BAG-1 and HSPA1A does not result in significant levels of refolded luciferase, and it may inhibit the activity of HSPA1A. (4) Overexpression of the mutated BAG-1 variants with HSPA1A resulted in no refolded luciferase similar to the WT BAG-1 protein with one exception. The overexpression of the variant M215V did not affect the refolding activities of HSPA1A. This finding suggests that the M215V mutation affects the BAG-1 function, maybe by altering the binding of BAG-1 to HSPA1A. The presence of an equal protein amount was verified using Western blotting (Appendix A).

## 3. Discussion

Based on the current databases and sequence similarity tools used, the *BAG1* gene is not present in bacteria or archaea. This finding strongly suggests that this protein is a eukaryotic innovation and is supported by the fact that all studied bacteria use GrpE as a nucleotide exchange factor [22].

Our results support homology (in contrast to analogy) between extant BAG-1 proteins and suggest that BAG-1 originated once early in eukaryotic evolution and subsequently was lost from a vast number of species. Our findings support an evolutionary scenario according to which the fusion between the UBQ-like and BAG domains occurred only once before the separation of the major eukaryotic kingdoms of animals, fungi, and plants. After the domain fusion event, the molecule followed separate and independent evolutionary trajectories in the different lineages, resulting in the very radical distribution in the extant animals, fungi, and plants for which genomic information is available.

This evolutionary pattern is consistent with the birth-and-death model of evolution [15,42], in which a particular gene as it passes vertically during speciation either duplicates (gene birth) or disappears (gene death). According to this model of evolution, the duplicated gene (paralog) is “free” from functional constraints and thus accumulates more mutations as compared to the original (first) gene copy [15,42]. The latter idea could explain the high divergence of BAG-1 proteins between different phyla, e.g., animals and plants, because the genes found in the extant species might not be direct orthologs. Instead, these sequences might represent paralogs that have been differentially lost independently in the different eukaryotic lineages. Alternatively (but not necessarily mutually exclusive), the low conservation can be explained by relaxation of purifying selection (that allows the accumulation of many amino acid-altering mutations as long as the major function–structure is preserved) or even by the function of positive (diversifying) selection, which favors amino acid altering mutations [43,44]. On the other end of the spectrum, however, we observed the high level of conservation of particular amino acids that seem to preserve the structure (three antiparallel α-helices) and NEF function (binding to the NBD region of Hsp70s) [24,39]. Although this conservation could be the result of genetic drift, in which particular mutations are randomly fixed in a population, we favor the function of purifying selection. Therefore, it seems probable that BAG-1 has evolved under a mixture of different evolutionary forces. Whether these forces resulted in different or specialized functions remains unknown, although the presence of several conserved residues of yet unknown functionality between all BAG-1 proteins studied and multiple phylum-specific conserved residues support this notion. Furthermore, it is not known which cellular functions attributed to BAG-1 other than being an NEF are conserved between the different species. Based on the observed conservation patterns in both domains, we speculate that some of these functions might be conserved. The association of BAG-1 with the proteasome observed in mammals [45] and plants [46] provides some support to this speculation.

If the ideas presented above stand true, would such mixed evolutionary patterns also be visible within a single species’ populations? Indeed, analysis of the type, frequency, and distribution of SNPs present within *BAG1* revealed evolutionary patterns similar to those observed between species. First, the finding that most SNPs within *BAG1* are rare suggests that *BAG1* follows a similar model of evolution within humans as it has between species. Second, the observation that SNPs found within *BAG1* were unequally distributed across different regions of the gene agrees with the finding that conserved amino acids within BAG-1 are scattered across the primary amino acid sequence and suggests that at least certain regions within *BAG1* are resilient to change. Furthermore, the bimodal distribution of amino acid-altering SNPs (nsSNPs) across BAG1’s coding region reveals differential accumulation of mutations within the two major functional domains (UBQ-like and BAG). These findings could be the result of diverse selective pressures acting on these two gene regions. Although the presence of different selection pressures acting upon the two regions (purifying on the UBQ and positive on the BAG) could explain the current findings, the random accumulation of mutations that have not been eliminated from the population cannot be ruled out. Together, these data suggest that SNPs occurring within BAG1’s coding region are subject to relaxed purifying selection, potentially due to less constrictive functional constraints, which could be explained partially by the redundancy within the Hsp70 networks. Whether or not this is the case, these suggestions are further corroborated by the interspecies evolution of BAG-1, which also revealed the presence of multiple radical amino acid changes.

To test some of the different evolutionary hypotheses, we characterized the functional outputs of a few selected SNPs. Based on the computational analysis, these mutations were not considered particularly radical in their chemical nature, although they were predicted to be “deleterious” by several algorithms. This assumption was partially verified by the almost identical melting temperature of all the recombinant proteins generated. On the other hand, some mutations altered the ATPase activity of HSPA1A differently from the wild-type BAG-1. However, it is very hard to determine or even pinpoint the mechanistic details of the observed changes because of the lack of any binding data. Moreover, if we assume that BAG-1 locks the HSPA1A in an “open” conformation [47,48], then the refolding assay suggests that the M215V mutation (which also affected Hsp70’s ATP hydrolysis) had lost this ability. This result could be related to changes in the packing interactions that resulted from the presence of valine in place of methionine. This finding could be interpreted as a loss-of-function of BAG-1 in homozygous individuals. Alternatively, these findings might suggest functional redundancy because other NEFs could compensate for a non-functional BAG-1 protein.

Collectively, the results of these assays suggest small but consistent changes in the function of BAG-1. This interpretation further suggests that the interaction of BAG-1 with Hsp70s does not depend on a single amino acid, but instead, it is a combination of smaller interactions that position the molecule in the correct orientation, thus allowing it to perform its NEF activity irrespective of the apparent affinity for the Hsp70 molecule [39]. Furthermore, we might not have observed major functional alterations because, except for the intracellular luciferase assay, the other in vitro assays did not contain other accessory molecules (e.g., other co-chaperones) [24,33,39,47,48,49]. Lastly, BAG-1 has several functions inside the cell, which include association with the proteasome, binding to Bcl2, and regulation of apoptosis [31,45], which were not tested in this report.

Evolutionarily, these observations suggest that functionally, BAG-1 is relatively plastic and can accommodate a high mutational weight without losing much of its original functionality. This speculation implies that most mutations during BAG-1’s evolutionary history within or between species are functionally and thus evolutionary neutral. Furthermore, the lack of major functional outcomes is less supportive of positive or diversifying selection in humans.

Sketching BAG1’s origin and evolution between several eukaryotes and within humans has revealed common evolutionary patterns that delineate the interspecies evolution and intraspecies variation of BAG-1. This knowledge provides insights into how BAG-1 has acquired its diverse functions and potentially predicts its ability to accommodate a plethora of amino acid changes within humans that could have allowed them to adapt to their dynamic environments or predispose specific individuals for particular diseases.

## 4. Materials and Methods

### 4.1. Origin and Evolution of BAG-1

To determine the evolutionary pathway and conservation of BAG-1 and its domains, various BAG-1 sequences were collected and compared using phylogenetic, sequence, and protein analysis tools.

#### 4.1.1. Sequence Collection and Alignment

Protein sequences were collected from the National Center for Biotechnology Information (NCBI) reference protein database (as of December 2019). The homologous sequences were identified using protein-BLAST searches using the human BAG-1 (NP_001165886.1), the *A. thaliana* BAG-1 (NP_200019.2), or the fission yeast BAG-1 (NP_596760.1) as queries. The searches’ parameters were: Max target sequence of 20,000, expected threshold of 10 (a parameter that describes the number of alignments expected by chance when searching a database of a particular size. The lower the *e*-value, the more “significant” the match), word size of 6, BLOSUM62 scoring matrix, gap costs of 11 for existence and 1 for extension, condition compositional score matrix adjustments, and masking low complexity regions. Protein sequences with the lowest *e*-values, query coverage greater than 80%, and similar domain organization as the query proteins were collected. The BLAST searches were performed using different species databases to ensure the collection of all available proteins, e.g., mammals were searched separately, followed by birds, then reptiles, and all other species with available genome sequences at NCBI.

Collected sequences were then aligned (all homologs, animals only, plants only, fungi only) with the Multiple Alignment using Fast Fourier Transform (MAFFT) program version 7 [37], using the E-INS-i strategy with a BLOSUM62 scoring matrix, a gap opening penalty of 1.53, and a 0.0 offset value. Sequence logos for each alignment were generated using WebLogo [38].

#### 4.1.2. Phylogeny Reconstruction

To ensure orthologous relationships of the collected sequences, we generated Neighbor-Joining phylogenetic trees. After analyzing and verifying all sequences’ relationships, 28 protein sequences were selected to be used in the final dataset as representative sequences (species). These sequences were used to generate a Maximum Likelihood tree using the JTT matrix-based model with discrete Gamma distribution [5 categories (+G, parameter = 5.6409)], and invariant sites (+I, 1.1976% sites). Branch lengths for each tree were measured based on the number of substitutions per site. Next to each branch, bootstrap values represent the percentage of replicate trees out of 1000 replicates that resulted in the observed clustering of taxa. Any positions containing gaps or missing data were removed, resulting in 167 positions in the final dataset. All evolutionary analyses were conducted in MEGA7 [50].

#### 4.1.3. Analysis of Introns and Genomic Organization

To verify orthology and support homology (versus analogy) between the selected sequences, the conservation of the intron position and phase was assessed. The intron position and phase for each collected *BAG1* gene was determined by using Spidey to align genomic sequences and coding sequences that were collected from GenBank [51]. Furthermore, to determine whether the synteny observed between *BAG1* and its neighbor genes in humans is conserved in other species, the genes flanking the *BAG1* loci were analyzed using GenBank’s genomic information.

#### 4.1.4. Determining BAG1’s Structural Characteristics

To characterize the structural effects of the evolutionary process and determine conserved and functionally and structurally important amino acids, structural predictions for BAG-1 homologs from *H. sapiens* (NP_001165886), *H. vulgaris* (XP_002160797.2), *A. thaliana* (NP_200019.2), *S. pombe* (NP_596760.1), and *H. marmoreus* (KYQ43628.1) were performed using SWISS-MODEL [52]. The target structures used for the modeling were 1HX1.pdb and 4HWI.pdb. Figures for each BAG-1 homolog were then prepared using PyMOL version 1.3 (Schrödinger, Inc. New York, NY, USA), with BAG-1’s functional amino acids highlighted in orange.

### 4.2. Intraspecies Comparisons and Microevolution of Human BAG1

To identify the evolutionary process in a short period of time and characterize the *BAG1* gene changes in humans, we collected and analyzed natural variants using bioinformatics and statistical tools.

#### 4.2.1. Collection and Analysis of SNPs

To investigate the microevolutionary processes of *BAG1*, its genomic coordinates and single nucleotide polymorphisms (SNPs) were collected from Ensembl’s 1000 Genomes browser (http://grch37.ensembl.org/Homo_sapiens/Info/Index) using all available data present within the 1000 Genomes database (phase 3). The data were filtered to include SNPs found only in the gene region and in the 1000 Genomes project. The patterns of SNP density within the *BAG1* gene and surrounding genes were also calculated by dividing the total number of SNPs within a given region by that region’s sequence length in base pairs (bp).

#### 4.2.2. Determination of Functional Amino Acids of BAG-1

Studies that reported mutagenesis experiments on amino acid of known function for BAG-1 and its homologs were collected. The collected studies were mined based on experiments that resulted in functional consequences, such as loss-of-function or elimination of binding to HSP70. Collected homologs were aligned with human BAG-1. BAG-1 amino acids with known functions were cross-referenced with non-synonymous SNPs reported in the 1000 Genomes database.

#### 4.2.3. Predictions of BAG1 Non-Synonymous SNPs

To prioritize the non-synonymous SNPs found on the *BAG1* gene and determine which mutations will be experimentally tested, five major criteria were used to best predict their functional effect [15]. First, the SNPs were based on whether they change an amino acid of known function because a mutation on a site of established function would almost most likely have a functional impact. Second, the SNPs were categorized based on whether they occur on a highly conserved amino acid position by determining the amino acid conservation level of each position because highly conserved amino acids would have a higher probability of inflicting a functional change. Third, the SNPs were sorted based on their frequency in a population. We aimed to identify mutations established in a population and that may be associated with major adaptations or are new and are related to a disease. Fourth, the SNPs were also classified based on whether the amino acid change was predicted to be radical (different amino acid class and negative or zero scores in both BLOSUM 65 and 80) [53]. This criterion’s rationale relies on the fact that radical changes may alter the function with a higher probability than non-radical amino acid changes. Fifth, we generated three-dimensional models of the WT and mutated proteins and predicted whether a mutation is altering the local conformation or the molecule surface.

### 4.3. Functional Characterization of Selected Mutations

To establish the effects of naturally occurring variants in the function of the BAG-1 in humans, we tested whether these mutations affect protein stability and function using biochemical and cell-based assays.

#### 4.3.1. Generation of Mutated Recombinant Clones, Proteins, and Protein Purification

The cDNA clone containing the *BAG1* gene sequence was amplified using PCR (see Table 7 for primers and [15] for conditions). Directional cloning of the purified PCR product corresponding to the BAG-1S protein isoform in both bacterial and mammalian expression vectors was performed as described in [15]. Site-directed mutagenesis using long-PCR amplification followed by DpnI digestion was used to generate the mutated *BAG1* variants [15,54]. After sequencing verification, the mutated and wild-type (WT) constructs were used to generate and purify recombinant proteins described in [55].

#### 4.3.2. Thermal Shift Assay

To explore the functional consequences of mutations on protein structure stability [15,54], we deployed the Thermal Shift Assay (Thermo Scientific; Waltham, MA, USA) to measure protein melting temperature (Tm). We used 5 µM of BAG1 WT and mutants, mixed with 5 µL of Protein Thermal Shift™ Buffer (Thermo Scientific; Waltham, MA, USA) followed by 2.5 µL of diluted Thermal Shift™ Dye (8X) for a 20-µL total volume. The mixture was incubated under continuous ramp mode in a CFX96^®^ Real-Time PCR Detection System (BioRad; Hercules, CA, USA) starting from 16 to 95 °C with a 0.05 °C/s rate. This experiment was conducted in triplicates, and the data were then plotted and fitted using the Boltzmann equation in SigmaPlot v10 (Systat Software Inc. San Jose, CA, USA) to determine the Tm. The mean and standard deviation of Tm values were calculated.

#### 4.3.3. ATPase Assay

A colorimetric assay was used to measure the free inorganic phosphate produced by hydrolysis of ATP from HspA1A in the presence or absence of two different amounts of BAG-1. Specifically, 1 µM of Hsp70 was incubated with 4 mM of ATP at 37 °C for 90 min in the presence or absence of 0.5 or 2 µM of BAG-1, and the inorganic phosphate (Pi) was measured every 30 min using a QuantiChrom^TM^ ATPase/GTPase Assay kit (BioAssay Systems; Hayward, CA, USA) [15,54,55]. The reactions were stopped using 100 µL of 34% citric acid [56]. A standard curve was generated by measuring the absorbance of Pi of known concentrations. Control groups containing no chaperones were used to determine the amount of spontaneous ATP hydrolysis. The data are presented as the difference of released Pi from time zero.

#### 4.3.4. Overexpression of BAG1 and Mutants in Mammalian Cells and Luciferase Protein Refolding

To determine whether any of the mutations affect the ability of Hsp70s to refold proteins, the luciferase refolding assay was used [54,57]. For this assay, we used HeLa cells, which were kept at 37 °C in a humidified 5% CO_2_ atmosphere. The cells were grown in a complete medium containing MEM, 10% fetal bovine serum, 2 mM L-glutamine, and penicillin-streptomycin.

The use of firefly luciferase provides stable and consistent results for the rate at which heat-denatured proteins will refold after stress. Luciferase reporter systems have long been used to assay the changes that occur due to alterations of molecular chaperone systems. They can provide insight into the importance of loss or changes of the chaperone function has on protein refolding in both prokaryotic and eukaryotic systems [57].

HeLa cells (ATCC, CCI-2) were split into 6-well plates at 2.0 × 10^6^ cells/well and grown in complete media (MEM, 10% NGS, P/S) for 24 h in standard cell culture conditions. Cells were then split into three labeled 24-well plates: 37 °C control, 45 °C with 0-min recovery, and 45 °C with 60-min recovery. Plates containing cells were transiently co-transfected with PolyJet In Vitro DNA Transfection Reagent (SignaGen; Frederick, MD, USA) using 1 µg/mL of total DNA (final concentration). Specifically, the cells were transfected with 0.5 µg/mL of firefly luciferase (Promega; Madison, WI, USA) and 0.5 µg/mL empty-GFP; 0.5 µg/mL of firefly luciferase and 0.5 µg/mL HSPA1A-GFP; 0.5 µg/mL of firefly luciferase and 0.5 µg/mL WT-BAG-1-GFP; and 0.33 µg/mL of firefly luciferase, 0.33 µg/mL HSPA1A-GFP, and 0.33 µg/mL of the WT or mutant BAG-1-GFP following the manufacturer’s protocol [15,54]. Transfection was allowed for 18 h, and then transfection media was removed and replaced with fresh complete media.

After six hours, media was removed and replaced once again with a 20-mM solution of 3-Morpholinopropane-1-sulfonic acid (MOPS; Life Technologies; Carlsbad, CA, USA) in fresh complete media. Cells were then incubated under standard culture conditions in the MOPS-containing media for 20 h, at which time the MOPS-containing media was removed and replaced with the “heat-shock buffer”, which contained a final concentration of 20 mM MOPS and 40 µg/mL cycloheximide (Sigma-Aldrich; St. Louis, MO), diluted in fresh complete media. Plates were placed back into the incubator for 30 min to stabilize cell lines to new media and inhibit protein synthesis. After the 30-min incubation, heat-shock plates were placed into a 45 °C water bath for 30 min to denature luciferase.

After stress, the cells in the 37 °C control and the 45 °C with 0-min recovery were resuspended through trypsinization and counted using the Cellometer^®^ Auto X4 Cell Counter (Nexcelom Bioscience, Manchester, UK). Cell counts were calculated and averaged to create a fixed concentration of cells for each construct. Resuspended cells were spun down (700× *g* for 2 min), and the media was aspirated. Cells were then resuspended in ice-cold 1x PBS to an 8 × 10^5^ cells/mL concentration, and 25 µL of cell suspension were aliquoted in triplicate into a white clear-bottomed 96-well plate. After cell suspension was added, a 1:1 ratio (25 µL) of Dual-Glo^®^ Luciferase reagent was added, the plate was gently swirled, and then incubated in the dark for 10 min. After incubation, luminescence was measured using a Promega GloMax^®^ luminometer (Promega; Madison, WI, USA).

The same procedure was then replicated for the 45 °C treatment with 60 min of recovery to obtain the recovery measurement. The raw luminescence measurements were averaged and normalized as percentages of the ratio of the experimental average to the GFP 37 °C control average, and the standard deviation was calculated. The averages for three independent experiments had their quartiles, means, medians, and standard deviations calculated and displayed as box plots. The presence of an equal protein amount was verified using total protein stain and Western blotting. Cell lysis and Western blot (including total protein stain) were performed exactly as described in [9,54] using an anti-GFP tag antibody (OmicsLink™ #CGAB-GFP-0050, Mouse Monoclonal IgG1 (Dilution 1: 1000) (Genecopoeia; Rockville, MD, USA).

### 4.4. Statistical Tests

One-way ANOVA with post-hoc Tukey HSD Test was used to determine statistical significance. In all cases, a *p*-value < 0.05 was considered statistically significant. The boxplots were generated using the R software (http://shiny.chemgrid.org/boxplotr/) [58].

## Figures and Tables

**Figure 1 ijms-21-09701-f001:**
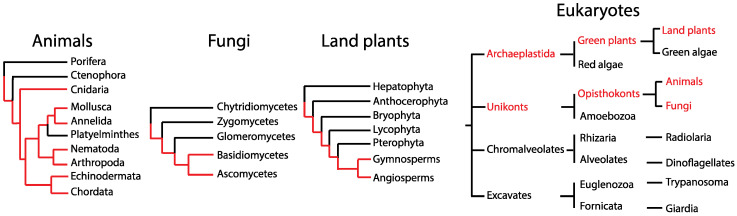
Presence of BAG-1 (red font) in eukaryotes based on BLAST results. Based on the parsimony criterion, BAG-1 may have originated before the major eukaryotic kingdoms split and was subsequently lost within individual taxonomic groups in each kingdom. Red lines represent the evolutionary history of the ancestral BAG-1. The relationships between the different taxonomic groups were obtained from the tree of life project (http://tolweb.org/Eukaryotes/3).

**Figure 2 ijms-21-09701-f002:**
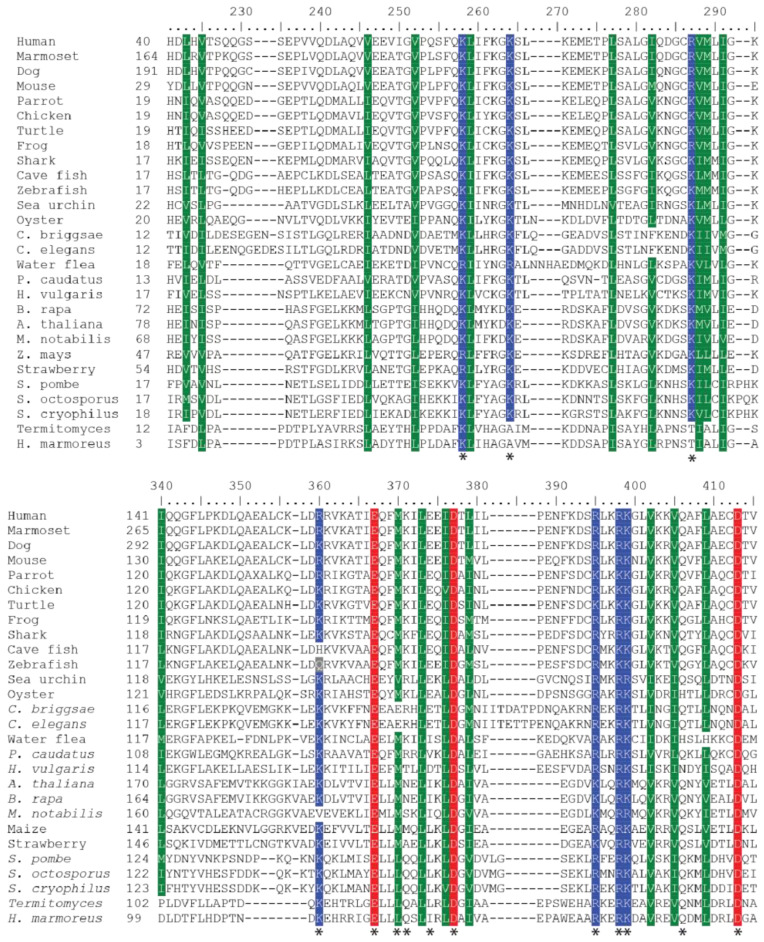
Divergent BAG-1 orthologs share similar functional amino acids. The multiple sequence alignment of the UBQ-like (top) and BAG domain (bottom) for 26 BAG-1 orthologs was generated using MAFFT version 7, with the E-INS-i strategy [37]. Residues labeled with an asterisk represent known functional amino acids present within the human BAG-1 [25].

**Figure 3 ijms-21-09701-f003:**
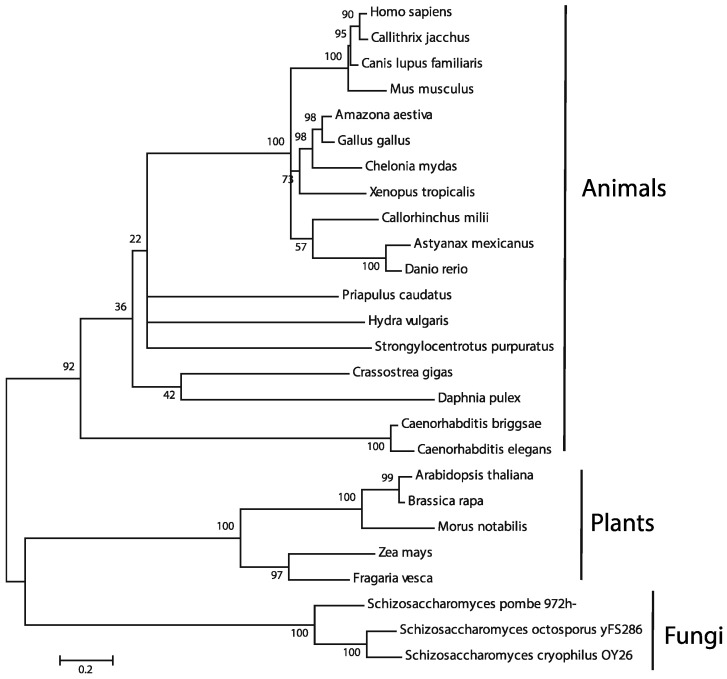
Phylogenetic reconstruction reveals clear orthology within each kingdom. Phylogenetic relationships between 26 BAG-1 homologs were determined using the Maximum Likelihood method based on the JTT matrix-based model, with complete deletion of amino acid positions containing gaps or incomplete datum. The tree with the highest log likelihood (−6107.0776) is shown. Bootstrap values indicate the percentage of 1000 replicates in which the associated taxa clustered in the resulting tree. The tree is drawn to scale, with branch lengths measured in the number of substitutions per site. The accession numbers of the sequences used, and the species’ scientific and common names are shown in Appendix A.

**Figure 4 ijms-21-09701-f004:**
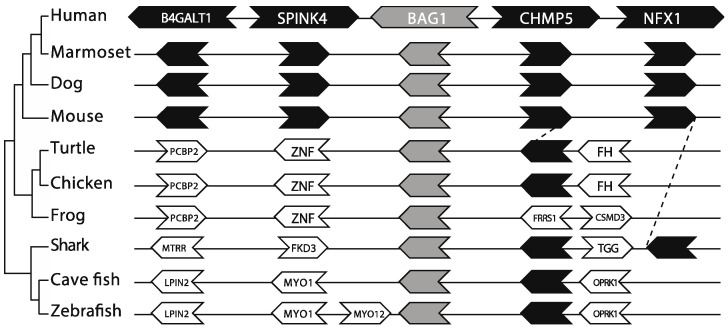
The genomic organization of the *BAG1* loci is relatively conserved within vertebrates. Genes flanking the *BAG1* loci were collected from GenBank. Filled arrows represent orthologs of genes flanking the human BAG1. Open arrows represent other genes surrounding the *BAG1* loci in each representative species. Each arrow’s direction represents the gene’s transcription orientation. The graph is not in scale for clarity.

**Figure 5 ijms-21-09701-f005:**
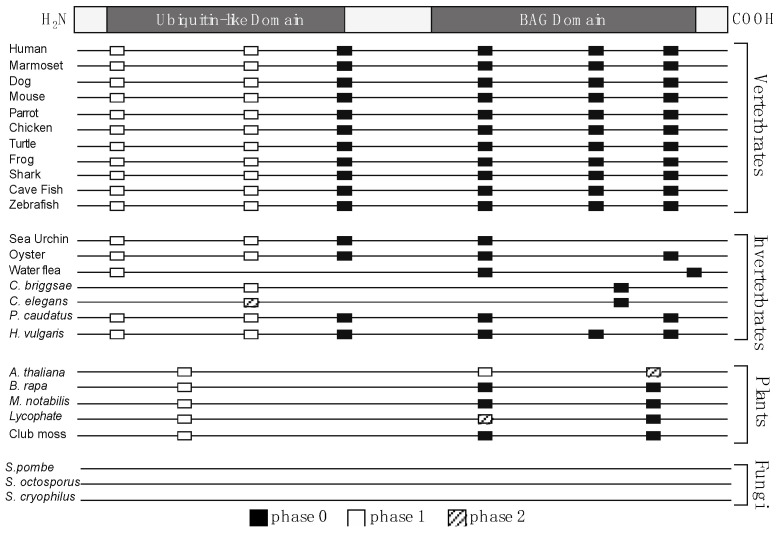
Conservation of intron position and phase reveals orthology within—and potentially between—animals, plants, and fungi. The figure depicts the position and phase of introns within *BAG1* genes from representative species in each kingdom. Intron position and phase were determined by aligning genomic sequences and coding sequences for each *BAG1* ortholog using Spidey (Wheelan et al., 2001).

**Figure 6 ijms-21-09701-f006:**
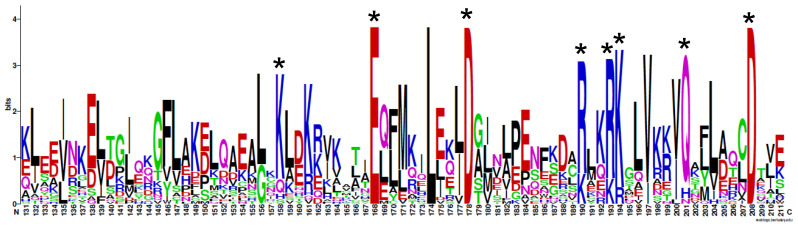
BAG-1 homologs share all the known Hsp70-interaction sites. Multiple sequence alignments were generated for BAG-1 homologs using MAFFT version 7. MAFFT outputs were then aligned to the human BAG1 and manually edited using BioEdit, to remove any gaps present within the BAG domain. Sequence logos were then generated for a particular region, including the BAG domain using the WebLogo server [38]. Positions marked with a black asterisk represent highly conserved amino acids required for Hsp70 interaction [25]. Sequence logos were generated using sequences from 129 animals, 32 plants, and 5 fungi species.

**Figure 7 ijms-21-09701-f007:**
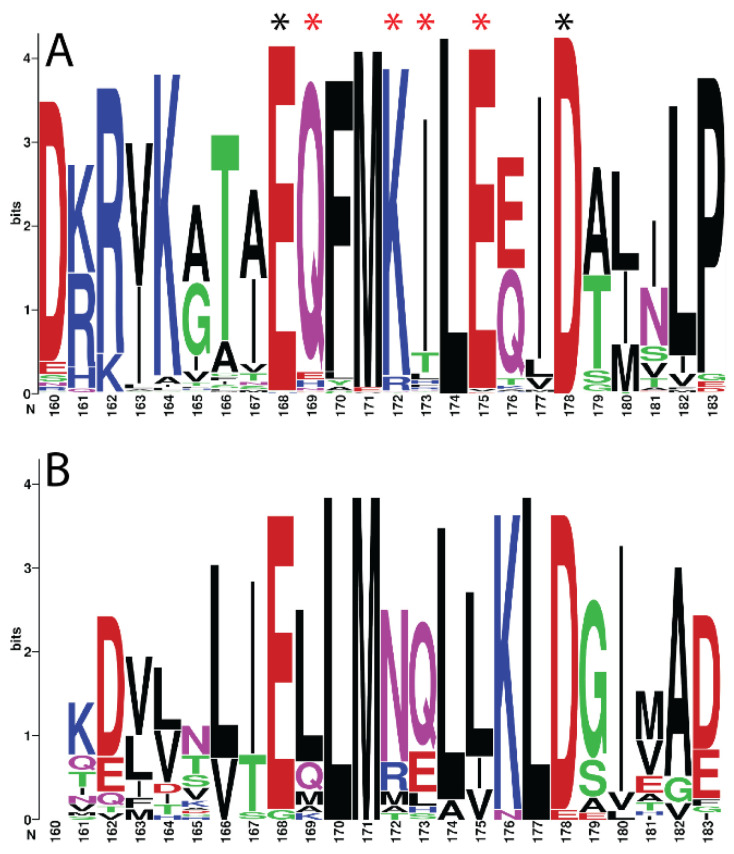
Sequence logos reveal signatures of functional divergence within each kingdom. Multiple sequence alignments were generated for BAG1 homologs from (**A**) animals, (**B**) plants, and (**C**) fungi. After editing (like we did for Figure 6) to remove gaps, sequence logos were generated for a particular region within the BAG domain (aa 160–183). Positions marked with a black asterisk represent highly conserved amino acids required for Hsp70 interaction [25]. Positions marked with a red asterisk represent conserved differences between kingdom-specific BAG1 homologs. Sequence logos were generated using sequences from 129 animals, 32 plants, and 5 fungi species.

**Figure 8 ijms-21-09701-f008:**
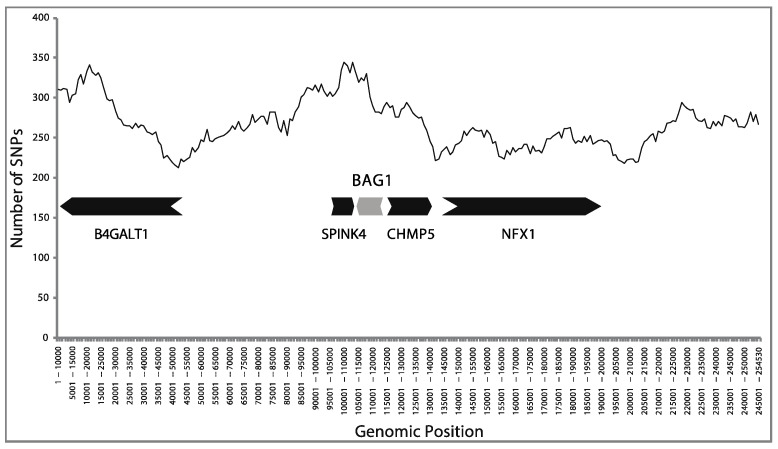
Distribution of the total SNP number of the *BAG1* locus genomic region. The plot was generated with an in—house script that plots the total number of SNPs in a window of 10,000 using a 1000 step size. The boxes represent the gene loci found in the 200-Kb region: *B4GALT1*, *SPINK4*, *BAG1*, *CHMP5*, *NFX1*.

**Figure 9 ijms-21-09701-f009:**
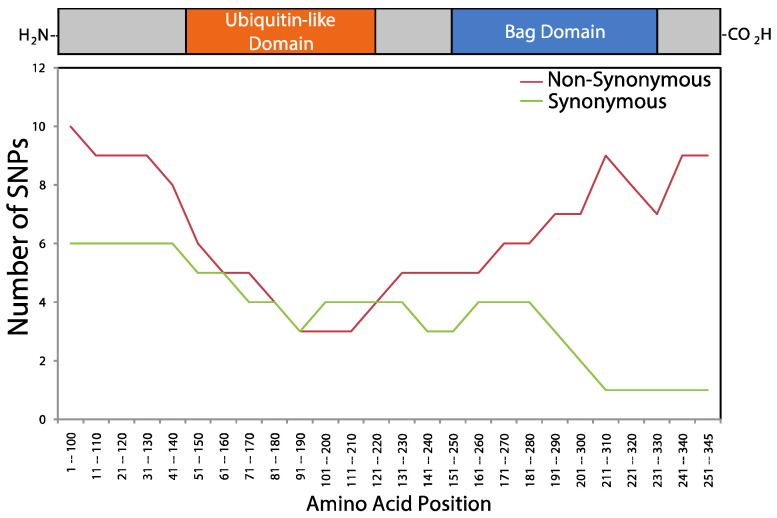
Non—random distribution of synonymous and non-synonymous SNPs within *BAG1*. SNPs were collected from Ensembl’s 1000 genomes browser. The frequency of SNPs across BAG1 was plotted using a window size of 100 amino acids and a step size of 10 amino acids.

**Figure 10 ijms-21-09701-f010:**
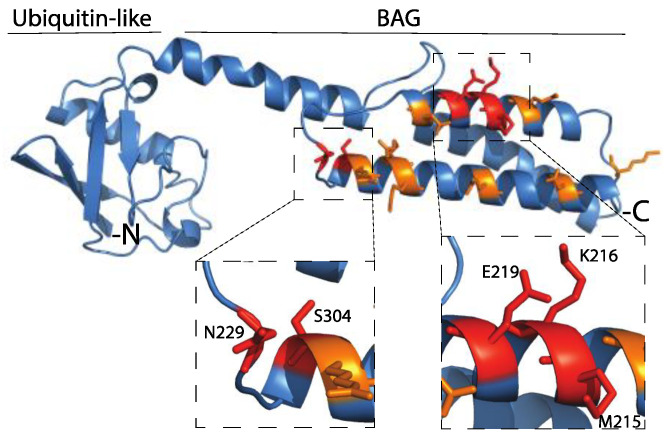
The human BAG-1 protein’s three-dimensional structure shows the amino acids’ position interacting with Hsp70 (orange) and the position of the naturally occurring mutations.

**Figure 11 ijms-21-09701-f011:**
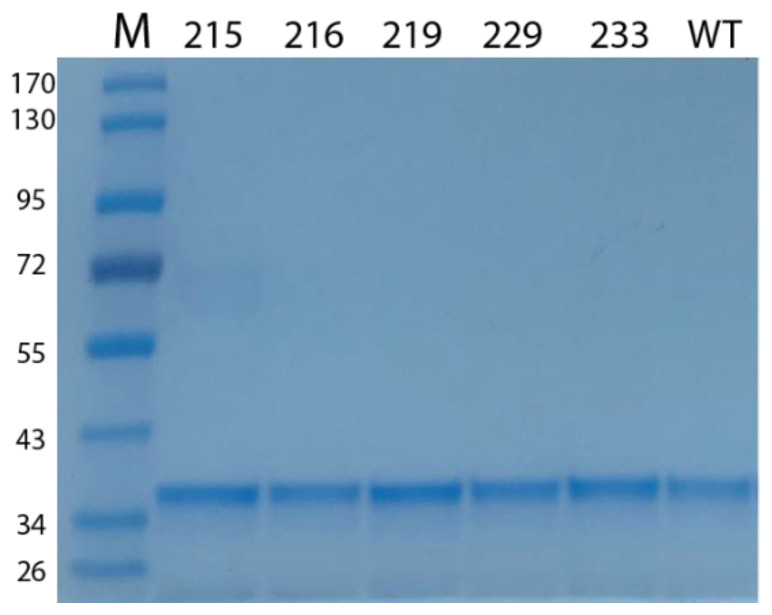
Purified recombinant proteins corresponding to the wild-type (WT) and mutated BAG-1 proteins. Approximately 1 μg of protein was loaded on an SDS-PAGE gel and stained with Coomassie blue stain M: molecular marker.

**Figure 12 ijms-21-09701-f012:**
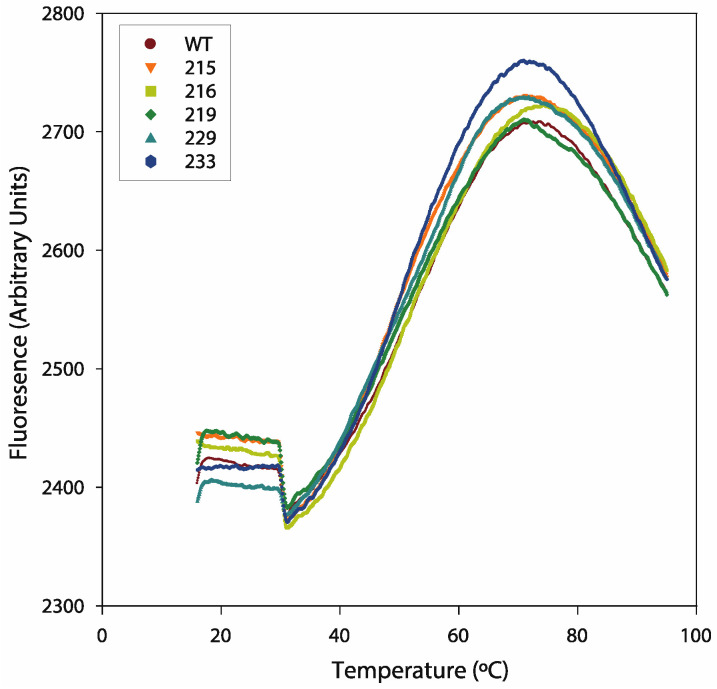
The thermal shifts produced by the recombinant proteins corresponding to the mutated BAG-1 variants were similar to those produced by the wild-type (WT) protein. Representative thermal shift assay data of a single batch of recombinant protein.

**Figure 13 ijms-21-09701-f013:**
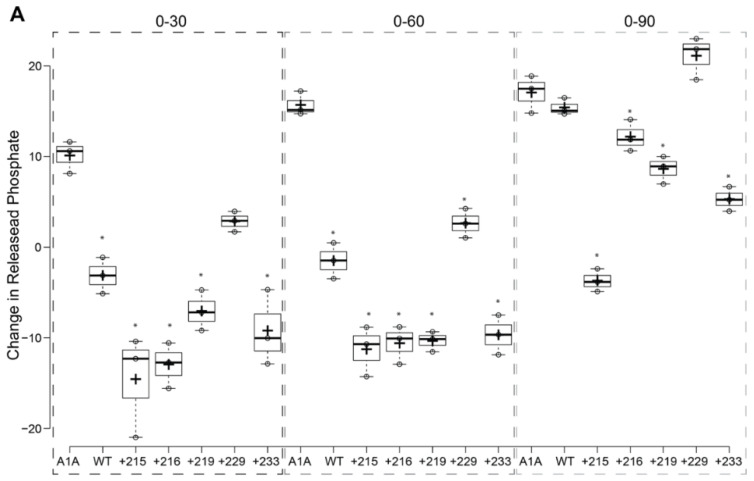
Wild type (WT) BAG-1 and its mutated variants alter in different ways the ATPase activity of recombinant HSPA1A. Two different ((**A**): 0.5 μM and (**B**): 2 μM) concentrations of BAG-1 (WT and mutants) were incubated with the same amount of HSPA1A (1 μΜ) in the presence of ATP. The data are presented as the change of released phosphate of three different time points (30, 60, and 90 min) from the released phosphate at time 0. Centerlines show the medians; box limits indicate the 25th and 75th percentiles as determined by R software; whiskers extend 1.5 times the interquartile range from the 25th and 75th percentiles; crosses represent sample means; dots represent individual data points. The Y-axes values are not the same between the two panels. Each experiment was repeated three times. Significantly different values are denoted with a star (*). The actual *p*-values between all comparisons are provided in Appendix A.

**Figure 14 ijms-21-09701-f014:**
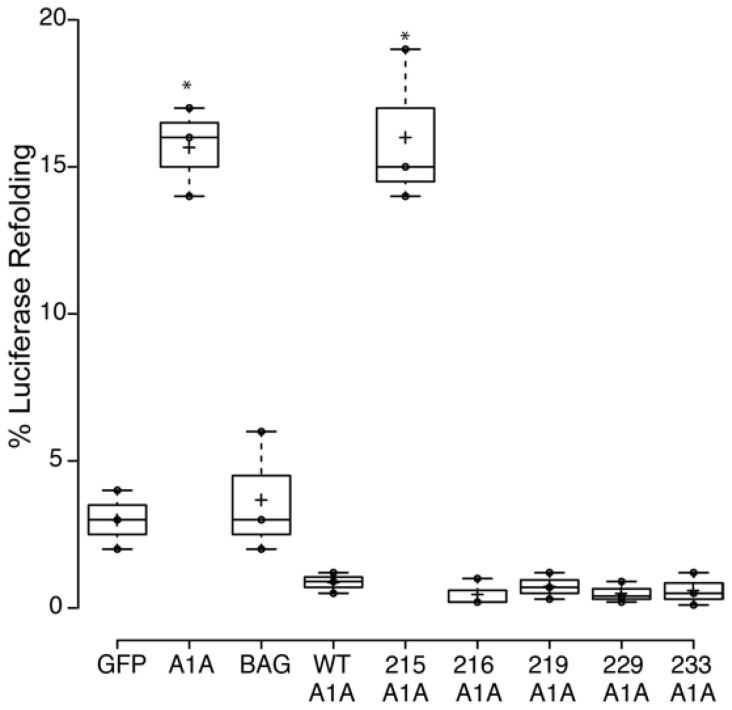
BAG-1 natural variants alter the refolding properties of HSPA1A (A1A). All mutations inhibit refolding of heat-denatured luciferase by HSPA1A at a level similar to the wild-type (WT) BAG-1 protein, except the K215E variants, which show significant changes as compared to the wild-type protein. Centerlines show the medians; box limits indicate the 25th and 75th percentiles as determined by R software; whiskers extend 1.5 times the interquartile range from the 25th and 75th percentiles; crosses represent sample means; dots represent individual data points. Each experiment was repeated three times. Significantly different values are denoted with a star (*). The actual *p*-values between all comparisons are provided in Appendix A.

**Table 1 ijms-21-09701-t001:** SNPs present within *BAG1*.

	Seq. Length (bp)	Number of SNPs	SNPs/1000 bp
Total	22,283	561	25.18
Regulatory	10,000	227	22.70
Genic	12,283	334	27.19
Intronic	8418	219	26.02
5’UTR	87	2	22.99
3’UTR	2740	76	27.74
Coding	1038	37	35.65
Syn.	245.17	12	48.95
Non-syn.	789.83	25	31.65

**Table 2 ijms-21-09701-t002:** Amount and location of rare and common SNPs within *BAG1 (synonymous: syn; non-synonymous: non-syn)*.

Population Frequency	Total	Regulatory	Genic	Coding	Syn.	Non-syn.	Deleterious
Rare	497	199	298	35	12	23	7
Common	64	28	36	2	0	2	0

**Table 3 ijms-21-09701-t003:** Distribution of coding SNPs within *BAG1*’s UBQ-like and the BAG.

SNP Type	Region	Length (bp)	Number of SNPs	SNPs/1000 bp
cSNPs	Full length	1035	37	35.75
	UBQ	231	6	25.97
	BAG	243	7	28.81
	Domain	474	13	27.43
	Non-Domain	561	24	42.78
sSNPs	Full length	244.83	12	49.01
	UBQ	52.83	3	56.79
	BAG	51	1	19.61
	Domain	103.83	4	38.52
	Non-Domain	141	8	56.74
nsSNPs	Full length	790.17	25	31.64
	UBQ	178.17	3	16.84
	BAG	192	6	31.25
	Domain	370.17	9	24.31
	Non-Domain	420	16	38.10

**Table 4 ijms-21-09701-t004:** Ratios of proportions of nsSNPs and sSNPs within BAG1.

Region	pN/pS
Full length	0.65
UBQ	0.30
BAG	1.59
Domain total	0.63
Non-Domain	0.67

**Table 5 ijms-21-09701-t005:** Properties of the selected amino acids used for functional assays.

dbSNP rs# Cluster id	Population	Global MAF	Amino Acid	Position in BAG-1M	Function	Conservation	SIFT Output	Polyphen Output
rs148464069	African ancestry (US)	0.0004	K/E	216	Hsp70 interaction	92.1%	deleterious	probably damaging
rs574806930	African (Gambia)	0.0004	M/V	215	Packing interaction	93.4%	deleterious	possibly damaging
rs114225550	African; American	0.0124	S/G	233	unknown	91.5%	deleterious	benign
COSM753616	pancreatic and lung cancer	0	E/K	219	Hsp70 interaction	93.4%	deleterious	probably damaging
COSM172723	Large Intestine cancer	0	N/I	229	Hsp70 interaction	86.1%	deleterious	probably damaging

**Table 6 ijms-21-09701-t006:** The melting temperature (Tm) of the wild-type and mutated BAG-1 variants were calculated by fitting the thermal shift assay data using the Boltzmann equation. The values are presented as the averages of three independent experiments. A student’s *t*-test determined statistical significance. A *p*-value < 0.05 was considered statistically significant. SD: standard deviation.

Sample	Tm (°C)	SD	N	*t*-Test (*p*-Value vs. WT)
WT	50.26	0.7852	3	na
K215E	50.64	0.9403	3	0.5514
M216V	50.63	0.7177	3	0.5068
S219G	51.13	1.0274	3	0.1508
E229K	48.49	0.6927	3	0.0148
N233I	51.62	1.2320	3	0.1124

**Table 7 ijms-21-09701-t007:** The sequences of the oligonucleotide primers used in the present study. The restriction enzymes sequences are capitalized, and the mutated nucleotides are underlined.

Name	Sequence (5–3)
5-NdeI-BAG1-pet	ccgCATATGatgaagaagaaaacccgg
3-XhoI-BAG1-pet	catCTCGAGctcggccagggcaaagtt
5-BAG1-XhoI-EGFP	ccgCTCGAGCatgaagaagaaaacccgg
3-BAG1-BamHI-EGFP	catGGATCCActcggccagggcaaagtt
5-BAG1-M215V	GCCACAATAGAGCAGTTTGTGAAGATCTTGGAGGAGATTG
3-BAG1-M215V	CAATCTCCTCCAAGATCTTCACAAACTGCTCTATTGTGGC
5-BAG1-K216A	CCACAATAGAGCAGTTTATGGAAATCTTGGAGGAGATTGACAC
3-BAG1-K216A	GTGTCAATCTCCTCCAAGATTTCCATAAACTGCTCTATTGTGG
5-BAG1-E219K	GAGCAGTTTATGAAGATCTTGAAAGAGATTGACACACTGATCCTG
3-BAG1-E219K	CAGGATCAGTGTGTCAATCTCTTTCAAGATCTTCATAAACTGCTC
5-BAG1-N229I	CACACTGATCCTGCCAGAAATTTTCAAAGACAGTAGATTG
3-BAG1-N229I	CAATCTACTGTCTTTGAAAATTTCTGGCAGGATCAGTGTG
5-BAG1-S233G	CTGCCAGAAAATTTCAAAGACGGCAGATTGAAAAGGAAAGGCTTG
3-BAG1-S233G	CAAGCCTTTCCTTTTCAATCTGCCGTCTTTGAAATTTTCTGGCAG

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
