# Peer review of "Origin and Evolution of the Human Bcl2-Associated Athanogene-1 (BAG-1)"

_ijms, 2020, doi:10.3390/ijms21249701_

Round 1

Reviewer 1 Report

In this work Nguyen et al set out to understand the evolution of the Hsp70 cochaperone, BAG-1. It is well known that Hsp70 utilizes nucleotide exchange factors to complete its chaperone mechanism. However, several nucleotide exchange factors are required in different organisms and their origins are unclear. Structurally these proteins exhibit some similarities though sequentially they diverge. Understanding the origin of this cochaperone can provide insight to chaperone mechanisms and perhaps shed light on other proteins that have co-evolutionarily evolved with it. The authors used the available sequence information and investigated its lineage, evolutionary relationships, genomic organization, and sequence conservation. They identified amino acids that were highly conserved and went on to test them experimentally.

The research question is clearly articulated, and the approach and results are explained well. However, I have a few questions and suggestions.

  1. In the introduction, DNAk and DNAJ should be changed to DnaK and DnaJ.
  2. Does BAG-1 have other known significant interactions other than Hsp70? Does it interact with Hsc70 similarly to Hsp70 or is this interaction isoform specific?
  3. There are multiple isoforms of some BAG-1 proteins that have deletions in various regions. How were these handled or accounted for in this work?
  4. This article is applicable to a broad range of people who may or may not have experience with sequence alignments. It would be useful to include a very brief discussion about the relevance of E-values, perhaps around the paragraph beginning at line 117 where it is touched on briefly.
  5. Figure 10 – what is the PDB ID of this structure?
  6. Is there any significance of residue 174? It is very interesting that it is very conserved but not involved in the interaction with Hsp70. Does BAG-1 have any other known function? This may also help to explain the lack of defects observed in the luciferase for many of the mutants.
  7. In the intracellular refolding assay where you are overexpressing Hsp70, luciferase, and the mutant BAG-1, how were the levels of the endogenous proteins handled?
  8. Supplementary S2 – please label the prominent bands. I assume the upper band corresponds to Hsp70. What is the lower band – I assume a control but it isn’t clear why there appears to be 2x the concentration in the control lane? Can you also show the amount of BAG-1?

Author Response

Reviewer 1:

In this work Nguyen et al set out to understand the evolution of the Hsp70 cochaperone, BAG-1. It is well known that Hsp70 utilizes nucleotide exchange factors to complete its chaperone mechanism. However, several nucleotide exchange factors are required in different organisms and their origins are unclear. Structurally these proteins exhibit some similarities though sequentially they diverge. Understanding the origin of this cochaperone can provide insight to chaperone mechanisms and perhaps shed light on other proteins that have co-evolutionarily evolved with it. The authors used the available sequence information and investigated its lineage, evolutionary relationships, genomic organization, and sequence conservation. They identified amino acids that were highly conserved and went on to test them experimentally.

The research question is clearly articulated, and the approach and results are explained well. However, I have a few questions and suggestions.

  1. In the introduction, DNAk and DNAJ should be changed to DnaK and DnaJ.

Done. Please see lines 71-74 in the revised manuscript.

  1. Does BAG-1 have other known significant interactions other than Hsp70? Does it interact with Hsc70 similarly to Hsp70 or is this interaction isoform specific?

This is great question. Based on our knowledge, BAG-1 interacts with both Hsp70 (HSPA1A) and Hsc70 (HSPA8). This assertion is supported by the literature (e.g., PMC1170124) and different interaction databases (summarized in NCBI’s GENE database: https://www.ncbi.nlm.nih.gov/gene?LinkName=protein_gene&from_uid=288915527under the tab “interactions”. We are not aware for any specific interactions between HSPA1A and HSPA8.

  1. There are multiple isoforms of some BAG-1 proteins that have deletions in various regions. How were these handled or accounted for in this work?

This is a great question that we did not properly addressed in the original manuscript.

BAG1encodes at least three protein isoforms, despite generating only one mRNA product (PMC1218990; PMID 9747877; PMID: 12902980). Three in-frame translation initiation sites account for the different protein isoforms. A non-canonical CUG start codon near the 5’ end of the transcript produces the 50 kDa, long isoform (BAG-1L). The first AUG start codon within the transcript produces the 46 kDa, medium isoform (BAG-1M). A second internal AUG start codon produces the 32 kDa, short isoform (BAG-1S) (PMC1218990; PMID 9747877; PMID: 12902980).

In this work, we used the BAG-1M in almost all sequence and phylogenetic analyses, because this conformation was the most conserved in the different species; thus, contained the most phylogenetic information. For the recombinant protein work and the luciferase assay, we used the shortest isoform BAG-1S. We also added some information on the isoforms in humans in the introduction. Please see lines 120-126, 380, and 650-651 in the revised manuscript.

  1. This article is applicable to a broad range of people who may or may not have experience with sequence alignments. It would be useful to include a very brief discussion about the relevance of E-values, perhaps around the paragraph beginning at line 117 where it is touched on briefly.

We thank the reviewer for the opportunity to further discuss E-values and their use in sequence identification. We briefly discuss the relevance of E-values in detecting protein homology in the revised manuscript. Please see lines: 142, 147, and 570-572 in the revised manuscript.

  1. Figure 10 – what is the PDB ID of this structure?

We apologize for omitting this information. We added the PDB codes used to generate the models in the materials and methods. Please see lines 606-607 in the revised manuscript.

  1. Is there any significance of residue 174? It is very interesting that it is very conserved but not involved in the interaction with Hsp70. Does BAG-1 have any other known function? This may also help to explain the lack of defects observed in the luciferase for many of the mutants.

This is an excellent point. We are not aware of a particular function for the residue 174 described in the literature. Indeed, BAG-1 has several functions that require the BAG domain (and in most cases ATP) and most probably depend on their interaction with Hsp70s. In the current manuscript, we focused on the specific function of BAG-1 as a nucleotide exchange factor of HSPA1A and did not test many of the other functions. The reviewer’s point allowed us to add a few sentences in the introduction and discussion revealing these functions raising the possibility that the mutations may affect any of these. Please see lines 119-125, 500-507, and 546-548 in the revised manuscript.

  1. In the intracellular refolding assay where you are overexpressing Hsp70, luciferase, and the mutant BAG-1, how were the levels of the endogenous proteins handled?

We did not alter the native chaperone content of the cells. Therefore, all the cells used in the current assays contained the same amount of endogenous proteins with the exception of the different overexpressed constructs.

We relied on the concept that heat-inactivated luciferase cannot refold by itself and the native (endogenous) Hsp70 systems are not enough in concentration to do so, but can refold in the presence of an overexpressed (excess amount) Hsp70 (PMID: 10909095; PMID: 30384997; PMID: 22297799). This concept is supported by the literature (please see citations above as examples) and the lack of refolding in the cells that did not express recombinant Hsp70 and overexpressed only empty GFP vector and BAG-1S-GFP construct.

  1. Supplementary S2 – please label the prominent bands. I assume the upper band corresponds to Hsp70. What is the lower band – I assume a control but it isn’t clear why there appears to be 2x the concentration in the control lane? Can you also show the amount of BAG-1?

We apologize for not labeling the blot properly. We have now labeled the bands.

The band closest to the top of the gel corresponds to the HSPA1A-GFP protein and the lower one corresponds to the BAG-1S-GFP protein. For further clarification, we also include the amounts of DNA used for transfection in the figure legend and the materials and methods (these were present in the material and methods in the original manuscript). Please see lines 664-668 and the legend of Supplementary Figure S2 in the revised manuscript.

The cells were transfected with 1 µg/ml total DNA. In particular, 0.5 µg/ml of firefly luciferase and 0.5 µg/ml WT-BAG-1-GFP; 0.33 µg/ml of firefly luciferase, 0.33 µg/ml HSPA1A-GFP, and 0.33 µg/ml of the WT or mutant BAG-1-GFP.

The amount of DNA used explains the higher concentration of recombinant protein present in the control (BAG-1S-WT alone) as compared to the lysates containing the experimental samples (BAG-1S and HSPA1A).

Unfortunately, we cannot show the endogenous BAG-1 amount, because we do not have an antibody against the endogenous protein. We relied on the antibodies against different tags. However, we theorize that it is not relevant to the conclusions, because all cells use had the same endogenous chaperone systems (please see relevant answer in comment 7).

Reviewer 2 Report

In the abstract the authors state that BAG-1 could have specialized and potentially unexplored functions during the evolutionary process. Which are exactly these functions?

Perhaps it would be the case to also introduce the role of HSPs in human physiology, for instance cardiovascular, immunity, or ecc., to give the readers an idea of the role of these proteins not only at intracellular level.

In the introduction, DNAk and further sentences lack references.

Are the results of figure 1 and 2 indicatively the same when considering DNA information as well in the BLAST?

Figure 11 has no western blot that usually complement the Coomassie staining.

The luciferase assays, in my opinion the only functional test present in the work, were performed in HeLa cells and are quite abiguous if one consider that cells do express endogenous BAG-1 protein. Can the authors normalize the data obtained, for instance in a western blot showing the recombinant and endogenous BAG-1 proteins?

Is there any GWAS study linking the K215E variant with human diseases?

Author Response

Reviewer 2:

  1. In the abstract, the authors state that BAG-1 could have specialized and potentially unexplored functions during the evolutionary process. Which are exactly these functions?

This is an excellent question. The short answer is that we do not know. The theoretical notion of specialized or unexplored functions stems from two major observations. First, BAG-1 and especially the BAG-domain region contains a number of highly conserved amino acid residues of yet unknown function in all species studied. Second, BAG-1 sequences between different phyla (i.e., animals, plants, and fungi) contain phylum specific conserved residues of unknown functionality. Although these observations support our assertion on functionality they do not provide enough room for further speculation as to what these functions might be.

To accommodate the reviewer’s comment, we first altered the abstract to reveal the uncertainty and second, added some information on additional BAG-1 functions in the introduction and discussed these in the light of our findings. Please see lines 119-125, 500-507, and 546-548 in the revised manuscript.

  1. Perhaps it would be the case to also introduce the role of HSPs in human physiology, for instance cardiovascular, immunity, or ecc., to give the readers an idea of the role of these proteins not only at intracellular level.

To accommodate the reviewer’s suggestion, we added some information in the revised manuscript. Please see lines 46-53 in the revised manuscript.

  1. In the introduction, DNAk and further sentences lack references.

We apologize for the omission, we added the references in the revised manuscript. Please see lines: 71-75 in the revised manuscript.

  1. Are the results of figure 1 and 2 indicatively the same when considering DNA information as well in the BLAST?

The results presented in Figure 1 are the summary of the observations from the BLAST searches using protein sequences. Figure 2 shows a representative multiple sequence alignment. The results will not change if DNA sequences used but many gene/proteins might be missed because of the lower conservation levels of the DNA sequences. We favor protein sequences to nucleotide ones, because of the higher level of conservation. Protein sequences will produce more robust results when studying long term evolution.

  1. Figure 11 has no western blot that usually complement the Coomassie staining.

The figure was included to show protein purity. We added the Western blot. Please see revised Figure 11.

  1. The luciferase assays, in my opinion the only functional test present in the work, were performed in HeLa cells and are quite abiguous if one consider that cells do express endogenous BAG-1 protein. Can the authors normalize the data obtained, for instance in a western blot showing the recombinant and endogenous BAG-1 proteins?

Unfortunately, we do not have experimental results of the endogenous BAG-1 levels, because we do not have an antibody against the endogenous protein. We relied on the antibodies against different tags. However, we theorize that it is not relevant to the conclusions, because all cells use had the same endogenous chaperone systems.

Specifically, the intracellular luciferase assay was performed as described in the literature (e.g., PMID: 10909095). We used HeLa cells, which do produce endogenous HSPA1A and probably BAG-1 (based on The Human Protein Atlas: www.proteinatlas.org). However, we have no credible indication that the endogenous proteins would alter the results, because in all transfections the background (endogenous) chaperones were the same and we used the same number of cells per experiment. Therefore, any observed changes are the result of the over-expressed proteins and not the endogenous. This notion is supported by two concepts. The first is a theoretical concept that heat-inactivated luciferase cannot refold by itself and the native (endogenous) Hsp70 systems are not enough in concentration to do so, but can refold in the presence of an overexpressed (excess amount) Hsp70 (e.g., PMID: 10909095; PMID: 30384997). The second is a practical concept that there is no luciferase refolding in the cells that did not express recombinant Hsp70 and overexpressed only empty GFP vector or BAG-1S-GFP construct.

For the above described reasons we do not believe we need to normalize the data using endogenous BAG-1 protein levels.

  1. Is there any GWAS study linking the K215E variant with human diseases?

This is an excellent question. Unfortunately, to our knowledge there are no GWAS studies that identified any of the residues stu

Round 2

Reviewer 2 Report

The authors replied to the raised questions.